# Description of the Fifth New Species of *Russula* subsect. *Maculatinae* from Pakistan Indicates Local Diversity Hotspot of Ectomycorrhizal Fungi in Southwestern Himalayas

**DOI:** 10.3390/life11070662

**Published:** 2021-07-06

**Authors:** Munazza Kiran, Miroslav Caboň, Dušan Senko, Abdul Nasir Khalid, Slavomír Adamčík

**Affiliations:** 1Department of Cryptogams, Institute of Botany, Plant Science and Biodiversity Centre, Slovak Academy of Sciences, Dúbravská Cesta 9, SK-84523 Bratislava, Slovakia; munazzakiran@gmail.com (M.K.); miroslav.cabon@savba.sk (M.C.); dusan.senko@savba.sk (D.S.); 2Institute of Botany, University of the Punjab, Quaid-e-Azam Campus, Lahore 54590, Pakistan; nasir.botany@pu.edu.pk

**Keywords:** agarics, Asia, bioclimatic zones, diversity hotspot, evolution, *Russulaceae*, phylogeny, vicariance

## Abstract

*Russula* subsect. *Maculatinae* is morphologically and phylogenetically well-defined lineage of ectomycorrhizal fungi associated with arctic, boreal, temperate and Mediterranean habitats of Northern Hemisphere. Based on phylogenetic distance among species, it seems that this group diversified relatively recently. *Russula ayubiana* sp. nov., described in this study, is the fifth in the group known from relatively small area of northern Pakistan situated in southwestern Himalayas. This is the highest known number of agaric lineage members from a single area in the world. This study uses available data about phylogeny, ecology, and climate to trace phylogenetic origin and ecological preferences of *Maculatinae* in southwestern Himalayas. Our results suggest that the area has been recently colonised by *Maculatinae* members migrating from various geographical areas and adapting to local conditions. We also discuss the perspectives and obstacles in research of biogeography and ecology, and we propose improvements that would facilitate the integration of ecological and biogeographical metadata from the future taxonomic studies of fungi in the region.

## 1. Introduction

*Russula* Pers. (Agaricomycotina, fungi) is an important genus of ectomycorrhizal (EcM) fungi with a high species diversity of which many are used as edible mushrooms in different parts of the world [1]. The research on *Russula* toxins and enzymes also shows a perspective in discoveries of compounds with potential medical and food industry uses [2]. It was hypothesized that previously described species from Europe and North America occur also in Asia [3,4,5]. Implementation of molecular techniques and use of ITS rDNA sequence as a barcoding tool proved that Asia has EcM taxa endemic to the continent [6]. Systematic research on some *Russula* lineages suggests existence of multiple diversity pools in the area of southern Asia [7], but diversity of northern boreal part of Asia probably in major parts overlaps with Europe [8]. Evidence regarding the endemic EcM species to southeastern and southern Asia boosted efforts to describe the local diversity. Therefore, multiple studies have recently emerged with new species from the area [9]. The current stage of Asian *Russula* research is exploratory, with a focus on species description and low attention to non-taxonomic reader. 

This study began with the detection of new genetic variability within *Russula* subsection *Maculatinae* (Konrad and Joss.) Romagn in Pakistan. The known diversity of *Russula* in Pakistan is overrepresented within this group and four species have already been described as new to science from relatively small area of southwestern Himalayas located in northern Pakistan. There are only seven other novel species of *Russula* described from Pakistan and supported by molecular phylogenetic analysis [9,10,11,12,13,14,15]. New collections from Pakistan with 97.75% similarity to closest published sequence in GenBank (MK966687 labelled as *R. dryadicola* originated from China) suggest that they may represent fifth *Maculatinae* species from this area. Recent studies have detected the highest number of known *Maculatinae* members in Europe with six species confirmed molecularly, from which four are agaric and two sequestrate species [16,17]. Southwest Himalayas in Pakistan seem to be the diversity hotspot of the group. Occurrence of high number of species indicates that there is a strong niche differentiation that acted as evolutionary driver and it also suggests that this is a possible ancestral area of the lineage [18]. However, low number of total *Russula* species published so far from western Himalaya (including Indian and Chinese territories) and locations of Pakistani *Maculatinae* records in a relatively small area suggest that the current knowledge of *Maculatinae* distribution in the Himalayas is incomplete and largely dependent on human factor. Publications describing *Maculatinae* members from Pakistan were facilitated by efficient morphological delimitation of the group and available European phylogenies [8].

In addition to standard taxonomic ambition to describe and circumscribe potentially new species of *Russula* subsection *Maculatinae* from Pakistan, the authors wanted to understand why so many species of the lineage occur on a relatively small area. To trace phylogenetic relationships and evolutionary history of the Pakistani *Maculatinae* we performed the multi-loci phylogenetic reconstruction. We analysed available ecological, climate and geographical data to understand niche specialization.

## 2. Materials and Methods

### 2.1. Sampling

The morphological description of new *Russula* member is based on our two collections from Khyber Pakhtunkhwa province. Morphologically they were assigned to *R. globispora* lineage based on the red pileus colour, brownish yellow spots on the stipe and yellow lamellae. They originate from moist temperate forests at altitudes above 2400 m dominated by *Abies pindrow* (Royle ex D.Don) Royle, *Cedrus deodora* (Roxb.) G.Don, *Pinus wallichiana* A.B. Jackson and with minor presence of evergreen broad leaf *Quercus* species. To understand niche specialization of Pakistani members of *Russula* subsection *Maculatinae*, we retrieved data about ecology and geographical position of already published *R. abbottabadensis* Saba and Adamčík [8], *R. mansehraensis* Saba, Caboň and Adamčík [19], *R. quercus-floribundae* Kiran and Adamčík [20] and *R. rubricolor* Jabeen, Naseer and Khalid [21].

### 2.2. DNA Extractions and Sequencing

Total genomic DNA was extracted from dried basidiomata using E.Z.N.A Fungal DNA Mini Kit (Omega Bio-Tek, Norcross, GA, USA) following the manufacturer’s recommendations. We targeted five DNA regions: (1) the internal transcribed spacer regions of nuclear ribosomal DNA (ITS); (2) partial large subunit of nuclear ribosomal DNA (LSU); (3) partial mitochondrial small subunit of ribosomal DNA (mtSSU); (4) the region between domains six and seven of the nuclear gene encoding the second largest subunit of RNA polymerase II (*rpb2*); (5) part of the translation elongation factor 1-alpha (*tef1α*). The following primer pairs were used for the DNA amplifications: ITS1F–ITS4 for ITS [22,23], LR0R–LR5 for LSU [24], MS1–MS2 for mtSSU [22], A-Russ-F–frpb2-7CR for *rpb2* [25,26], EF1-983F–EF1-1567R for *tef1α* [27]. All molecular markers were amplified with 5× HOT FIREPol^®^ Blend Master Mix (Solis BioDyne, Tartu, Estonia) applying PCR conditions of Caboň et al. [26]. The PCR products were purified using Exo-Sap enzymes (Thermo Fisher Scientific, Wilmington, Germany) and sequenced at SeqMe sequencing company (Dobříš, Czech Republic).

### 2.3. Sequence Alignment and Phylogenetic Analyses

Sequences were edited in the BioEdit Sequence Alignment Editor version 7.2.5 [28] or Geneious version R10 [29]. Intra-individual polymorphic sites having more than one signal were marked with IUPAC ambiguity codes (https://www.bioinformatics.org/sms/iupac.html, accessed on 31 May 2021). For phylogenetic analyses, our sequence data were supplemented by available sequences of published species from Pakistan (for sources see references in Sampling), other Asian regions [9,30], Europe [8] and North America [31]. Members of *Cuprea* lineage were selected as an outgroup following Caboň et al. [8]. We searched GenBank (https://www.ncbi.nlm.nih.gov/genbank, accessed on 31 May 2021) and UNITE (https://unite.ut.ee, accessed on 31 May 2021) databases for all available ITS sequences with Asian origin and with 97% similarity to closest *Maculatinae* member. Sequences used in this multi-loci phylogenetic analysis are listed in the Appendix A. Partial alignments of individual loci were aligned by MAFFT version 7 using the strategy E-INS-i [32], further manually improved and concatenated into single multi-loci dataset using SeaView version 4.5.1 [33].

Phylogenetic analyses were computed using the CIPRES Science Gateway [34]. The topology of ITS tree was not congruent with partial trees of other genes. As ITS, as a fungal barcode, is the most represented in public databases, the analysis of this region was run separately. The final multi-loci dataset of LSU, mtSSU, *rpb2* and *tef1α* was analysed with two methods: Bayesian Inference (BI) and Maximum Likelihood (ML). For the ML analysis, the concatenated alignment was uploaded as FASTA files and analysed using RAXML-HPC2 on XSEDE (8.2.12) [35] as a partitioned dataset under the GTR + GAMMA model with 1000 bootstrap iterations. For the BI analysis, the dataset was divided into seven partitions: nrLSU, mtSSU, intronic positions of tef1α, and the 1st + 2nd and 3rd codon positions of *rpb2* and *tef1α*. The best substitution model for each partition was computed jointly in PartitionFinder 1.1.1 [36]. The aligned FASTA dataset was converted to nexus format using Mesquite 3.61 [37] and further analysed using MrBayes 3.2.6 [38] on XSEDE (8.2.12) [35]. Bayesian runs were computed independently twice with four MCMC chains for 10 million generations until the standard deviation of split frequencies fell below the 0.01 threshold. The convergence of runs was visually assessed using the trace function in Tracer 1.6 [39]. The ITS dataset was analysed as a single partition by ML under the same settings as the multi-loci analysis. The resulting trees were visualized and annotated by TreeGraph 2 [40] and graphically improved in CorelDRAW X5 (Ottawa, ON, Canada).

### 2.4. Morphological Observations

Macromorphological characters were observed on fresh basidiomata. Microscopic characters were studied on dried specimens. Most of the microscopic objects (except of spores) were measured in Congo red solution with ammonia after a short treatment in aqueous 10% KOH under Olympus BX-41 microscope (Olympus, Japan) using an eyepiece micrometer and at magnification of 1000×. Spores were scanned with an Artray Artcam 300MI camera and measured by Quick Micro Photo (version 2.1) software. Enlarged scanned pictures of spores were used for measuring with an accuracy of 0.1 μm and for making line drawings. All other elements are measured with an accuracy of 0.5 μm. All drawings of microscopic structures, with the exception of basidiospores, were made with the aid of a camera lucida (Olympus U-DA, Olympus, Japan) at a projection scale of 2000×. Morphological description, terminology and used chemical reactions follow morphological *Russula* standards proposed by Adamčík et al. [9]. Statistics of microscopic dimensions are based on 30 or 20 measurements per sampled collection and given as a mean value (in italics) plus/minus standard deviation; values in parentheses give measured minimum or maximum values.

### 2.5. Analyses of Ecological, Geographical and Climate Data

Data regarding the geographical position of known *Maculatinae* samples from Pakistan, presence of associated tree species (as potential symbiotic partners) and other collecting data are summarised in the Table 1. Based on geographical position, we assigned global environmental zone and biome to each collection following high resolution bioclimate map of the world [41]. This map is built on 42 bioclimatic variables and has 30 arcsec resolutions (equivalent to 0.86 km^2^ at the equator). The map segment of northern Pakistan was digitalised and further edited in GRASS GIS 7.8.5 Geographic Resources Analysis Support System (GRASS) of the Geographic Information System (GIS) environment version 7.8.5, which was released under the GNU/GPL license. Global environments are classified into 7 biomes, 18 global environmental zones and 125 global environmental strata (GEnS) following Metzer et al. [41].

## 3. Results

### 3.1. Phylogeny

ITS tree (Figure 1) showed that the studied Pakistani collections formed a well-supported clade within *R. globispora* lineage. However, the topology of this lineage was in a large part polytomy showing only good support at some terminal nodes, from which at least some apparently correspond to the already described species. All Pakistani species received support: *R. abbottabadensis*, *R. rubricolor* and *R. quercus-floribundae* together with the two our Pakistani collections are all member of *R. globispora* lineage, while *R. mansehraensis* belonged to *R. maculata* lineage. Other Asian sequences from public databases did not form a supported cluster with any clade of Pakistani species. Collections of *R. globispora* lineage from southeastern Asia included *R. heilongjiangensis* G.J. Li and R.L. Zhao, *R. tengii* G.J. Li and H.A. Wen, two other undescribed species clades and a few singleton collections from Korea, Japan and China. The only Middle East sequence retrieved from GenBank was from Turkey and was placed close to European *R. globispora* (ML = 90). Within *R. maculata* lineage, Asian collections formed a monophyletic clade (ML = 92) sister to European members (ML = 95) with support for additional species from China and other singleton sequences from both China and Papua New Guinea.

Multi-loci analysis of LSU, mtSSU, *rpb2* and *tef1α* confirmed that studied Pakistani collections form a strongly supported clade within *R. globispora* lineage and are described in this study as *R. ayubiana* sp. nov. (Figure 2). The tree strongly supported relationship of the new species with alpine-arctic species *R. dryadicola* R. Fellner and Landa and *R. tengii*. This clade was further supported as sister to sequestrate *R. mattiroloana* (Cavara) T. Lebel. Other two Pakistani species, *R. abbottabadensis* and *R. quercus-floribundae* formed a well-suported clade. *R. mansehraensis* was confirmed to be a member of *R. maculata* lineage. The multi-loci phylogeny demonstrated existence of at least three lineages with independent origin in Pakistan.

### 3.2. Ecological, Geographical and Climate Data Analysis of Maculatinae in Pakistan

All molecularly identified *Maculatinae* members from Pakistan are represented by 5 species and altogether 16 collections (2–5 per species) collected during a time span of 7 years (2012–2018). All collections are situated in relatively small area with the most distant ones less than 200 km apart, all are from Khyber Pakhtunkhwa province and they are located in four Pakistani districts (Table 1). Their altitude range from 1109 to 2472 m, but apparently altitude did not limit distribution of at least two species, i.e., *R. rubricolor* and *R. mansehraensis* were both collected below 1500 m and also above 2000 m. *Maculatinae* collections were located in six geographical environmental strata (GEnS) that are further classified to four geographical environmental zones and two biomes (Figure 3). Three species; *R. abbottabadensis*, *R. ayubiana* and *R. quercus-floribundae*; are only present in single GEnS but this is due to low dispersal of collections that originate from single or a couple of collecting areas as different mycelia or different time duplicates (same mycelium origin collected at different dates). *Russula mansehraensis* and *R. rubricolor* have more dispersed collections that were each located in two to three GEnS, respectively, both were collected in two zones and the first also in two biomes. To our knowledge, *R. abottabadensis* and *R. mansehraensis* were clearly associated with pines (*Pinus roxburghii* Sarg.) and *R. quercus-floribundae* with oak (*Quercus floribunda* Lindl. ex Camus). It was impossible to decide about host tree of *R. ayubiana* because of mixture of different trees present at collecting site and authors of *R. rubricolor* also reported both conifer and deciduous tree as associated trees.

### 3.3. Taxonomy

***Russula ayubiana*** Kiran & Khalid sp. nov. Mycobank no.: 840084*Holotype:* Pakistan: Khyber Pakhtunkhwa province, Abbottabad district, Kuzagali, alt. 2430 m, in a mixed forest dominated by conifers, on moist, humus rich, acidic soil. 3 Aug 2016, Abdul Nasir Khalid *KG9* (LAH 35439).*Etymology*: the epithet “ayubiana” refers to the Ayubia National Park which is adjacent to the place (Kuzagali) where the holotype was collected.

Pileus (Figure 4) small to medium-sized, 60–90 mm diam., plano-convex with a slightly depressed centre, becoming applanate when old; margin deflexed, smooth, tuberculate-striate; cuticle smooth, wet-viscous and shiny, often with attached particles of debris, pruinose towards the centre, near the margin brownish pink to red, in the centre dark red and sometimes discolouring to pale olive. Lamellae moderately distant, equal, adnate-emarginate, pale yellow; lamellulae absent, furcations rare near the stipe; edge even, concoloured. Stipe 50–75 × 12–15 mm, obclavate to almost cylindrical, longitudinally striate, pruinose, white, with brownish yellow spots especially near the base. Context white, yellowish brown upon bruising, compact. Spore print not observed. 

Spores (Figure 5) (7.1–)7.8–*8.7*–9.7(–11.3) × (5.6–)6.5–*7.3*–8.1(–9.4) μm, broadly ellipsoid, Q = (1.08–)1.13–*1.2*–1.27(–1.41); ornamentation of large, prominent, distant to moderately distant [3–5(–6) in a 3 μm diam. circle] amyloid warts or spines, (1.3–)1.6–2(–2.4) μm high, occasionally fused (0–2 fusions in the circle) and connected by short line connections [0–1(–3) line connections in the circle] forming a short or a long chains that are radially oriented from the suprahilar spot), isolated spines occasional; suprahilar spot large, amyloid, finely warted and with irregular radial amyloid projections. Basidia (23–)28.5–*36.3*–44(–53) × (10–)12.5–*13.9*–15(–16.5) μm, broadly clavate, 2–4-spored; basidiola first cylindrical, then clavate, ca. 7–12 μm wide. Hymenial cystidia widely dispersed, ca. 150–200/mm^2^, (51–)66–*77.5*–88.5(–110) × (11–)13–*14.5*–16(–18) μm, fusiform, clavate or lanceolate, apically mainly acute, usually with a (2.5–)6–*10*–13(–15) μm long appendage, thin-walled; contents heteromorphous in almost 3/4 of the volume, granulose-crystalline or banded, apically more dense, slowly turning red-brown in sulfovanillin; near the lamellae edges more numerous, similar in size and shape, (44–)62–*75.5*–90(–103) × (8–)11–*13.5*–16(–18) μm, usually without an appendage. Lamellae edges fertile; marginal cells undifferentiated. Pileipellis weakly metachromatic in Cresyl Blue, sharply delimited from the underlying context, 95–112 μm deep, strongly gelatinized throughout; vaguely divided in 45–65 μm deep suprapellis of ascending, loose hyphal terminations, forming often pyramidal structures composed mainly of emerging fascicles of pileocystidia, covered with 30–70 μm deep, transparent gelatinous layer; gradually passing to 50–60 μm deep subpellis of denser, irregularly and near context horizontally oriented, intricate, 2–3 μm wide hyphae. Acid-resistant incrustations absent. Hyphal terminations near the pileus margin not or only slightly flexuous, composed of 1–3 unbranched cells, thin-walled; terminal cells (20.5–)24.5–*36.3*–48(–72.5) × (2–)2.5–*2.7*–3(–3.5) μm, subulate or cylindrical and apically attenuated or constricted, subterminal cells usually unbranched, often shorter or longer compare to terminal cells. Hyphal terminations near the pileus centre narrower and more frequently branched, 15.5–*27.9*–40(–84) × (2–)2.5–*3.2*–4(–5.5) μm, more frequently cylindrical and apically not constricted, subterminal cells more frequently branched. Pileocystidia near the pileus margin very abundant, in small fascicules, (1–)2–4(–6)-celled, cylindrical or subcylindrical and not distinctly narrowing towards the bases, thin-walled, terminal cells (24–)30–*41.9*–54(–71.5) × (2.5–)3–*4*–4.5(–6) µm, mainly cylindrical, occasionally narrowly clavate, apically mainly obtuse, contents heteromorphous, mainly crystalline or banded, occasionally also granulose, weakly turning greyish in sulfovanillin. Pileocystidia near the pileus centre similar, terminal cells (18.5–)21–*36.9*–52.5(–11.5) × (2–)3–*4.1*–5(–7) µm. Cystidioid hyphae in subpellis and context present, narrow (ca. 3 μm wide). 

*Additional material studied:* Pakistan: Khyber Pakhtunkhwa province, Abbottabad district, Kuzagali, alt. 2430 m, in a mixed forest dominated by conifers, 3 August 2016, Abdul Nasir Khalid *KG7* (LAH35438).

*Note: Russula ayubiana* has field appearance typical for *Maculatinae*: red pileus cuticle, pale yellow lamellae (suggesting that the spore print is yellow) and the surface is covered by brownish yellow spots. The spores are clearly smaller (in average shorter than 9 µm) compared with all known members of *R. globispora* lineage and in size they match rather *R. maculata* lineage [42]. The new species differs from *R. maculata* Quél. & Roze and its relatives in spores with relatively dispersed line connections and more prominent elements of the ornamentation. Pileocystidia have frequently three and more cells that is another character distinguishing it from the majority of known species of *R. globispora* lineage and shared with *R. tengii* and *R. quercus-floribundae* [8,20].

## 4. Discussion

Our study clearly demonstrated that all five Pakistani *Maculatinae* species are only known to be located in the southwestern Himalayas and belong to at least three independent lineages. Sequences of Pakistani *Maculatinae* from other foreign regions are not represented in public databases and none *Maculatinae* members are reported in the Russulaceae list from adjacent Indian region of Uttarakhand located in southwestern Himalayas [43]. No evidence of long dispersal of Pakistani *Maculatinae* members was detected that is similar to broad distribution of boreal-arctic species of Northern Hemisphere [44,45]. The results suggest that the area is a unique biodiversity region for the genus *Russula*; different from southeastern Asia, northern Eurasia and Europe. 

According to analyses of global metadata associated with publicly available ITS sequences, the genus *Russula* evolved ca. 55 million years ago (Mya) in temperate areas and *Maculatinae* belongs to the crown clade that splitted at ca. 44 Mya [44]. The individual species origins are estimated for an average of 3.3 Mya for extratropical taxa. When looking into details, the species relations and their age might be closer, e.g., age of *Russula* subsection *Roseinae* shows approximately similar branch length and phylogenetic divergence to *Maculatinae* that was estimated to 12.2 Mya, and the age of crown *Roseinae* clade that may correspond to *R. globispora* lineage was estimated to 6.96 Mya, with the majority of species younger than 2.6 and evolved in Pliocene [46].

All known *Maculatinae* collections from Pakistan are located in Himalaya biodiversity hotspot (https://www.conservation.org/priorities/biodiversity-hotspots, accessed on 3 May 2021). Biological diversity of the south Himalayas corresponding to this hotspot was formed by five major geophysical upheavals from Indian subcontinent collision at 45 Mya until the beginning of the quaternary climate fluctuations at 2.6 Mya [47]. Southwestern part of Himalayas was specifically influenced by climate drying and cooling in central Asia at 8–7 Mya associated with change of woodland to savannah and this resulted to decreased plant species diversity compare to southeastern Himalayas [48,49]. Sufficient data for molecular dating are not available, but if taking *Russula* subsection *Roseinae* Sarnari as an equivalent, it is most likely that *Maculatinae* migrated to southwestern Himalayas in several lineages after aridification of central Asia and before quaternary climate fluctuations between 8 to 2.6 Mya. Contemporary events in this period may support the hypothesis of host tree switch and expansion as driving *Russula* diversification [44].

Available data about Pakistani species are very limited in current form and do allow very few conclusions about ecological characteristics or niche specification. *Russula ayubiana* is the only species showing clear link between phylogenetic history and its current niche. The species is nested in a clade with two boreal-arctic-alpine species *R. dryadicola* and *R. tengii* (Figure 2). The new Pakistani species was collected at higher altitudes without apparent preference for host tree that roughly correspond to ecological characterisation of other two Asian species [8]. *Russula abbottabadensis* and *R. quercus-floribundae* are placed on sister positions, but they sit on long branches, which suggests that the resulting support on higher note may be result of a long-branch attraction. Including more Asian samples represented now only in ITS data may eliminate the heterotachy and reveal further relationships within this clade [50]. *Russula abbottabadensis* and *R. mansehraensis* both form apparently mycorrhiza with *Pinus roxburghii* and they are vicariant together with *R. rubricolor.* Two collections of *R. quercus-floribundae* associated with *Quercus floribunda* also do not serve as sufficient sampling to decide about niche differences from other Pakistani *Maculatinae* members. Our analysis of bioclimatic zones revealed vicariance and broad ecological amplitude of at least a part of *Maculatinae* members in western Himalayas. This can be explained as a result of adaptation to specific ecological conditions at a small, local scale [51]. Ecological function of ectomycorrhizal fungi resulting from their enzymatic activities involved in symbiotic interaction with trees also play an important role in shaping fungal communities [52]. However, to understand speciation processes, we need to focus on local adaptation and gene flow rather than distribution pattern and coexistence of species [53]. Important factor that shapes speciation processes in *Russula* is a fundamental function of symbiosis [54]. Above we discussed how much we could understand from the available data about evolutionary processes, ecological adaptations, niche specialisations, speciation drivers, etc. Our data are not sufficient to make conclusions. The structure of recent publications describing new *Russula* species from Asia is often simple and straightforward: they describe a single or a pair of species based on ITS sequence data, usually defined by unique and distinct morphological characters. Species are picked opportunistically from different unrelated groups. There are only a few very recent publications using more DNA loci (including single-copy genes) describing new Asian *Russula* species within a lineage of closely related species; for example Wang et al. [55] described two new species of *Russula* subsect. *Substriatinae* X.H. Wang and Buyck from China, Wisitrassameewong et al. [7] described three species in the *Russula* sect. *Amoeninae* Buyck and Caboň et al. [8] described three species of *R. globispora* lineage from Asia. The latter study includes *R. abbottabadensis* from Pakistan and the data become a solid base of other species descriptions and are well-implemented in our study. Such publications facilitate further research of the diversity of *Russula* lineages globally. 

More collections data about wild fungi are required to understand their commercial use as addition to human diet and their conservation status. Little is known about edible fungi in Pakistan [56]. According to our communication with local people, few *Russula* species are collected for consumption but there is not a single published report about any edible *Russula* sp. in the country [57,58]. *Russula* is apparently among the most valuable traded edible fungi in China [59] and also in Himalayan regions of Northern India and Nepal, but they are usually members of subgenera *Heterophyllidiae* Romagn. and *Brevipedum* Buyck and V. Hofst. [60,61]. We did not find any evidence about *Maculatinae* used as edible fungi globally [58,62]. In Europe, many Russulas are collected as edible based on mild taste of lamellae context (https://www.wildfooduk.com/articles/identifying-russulas/, accessed on 4 May 2021) and if applying this taste test on *Maculatinae* of Pakistan, probably all species of *R. globispora* lineage will qualify as edible. However, this taste testing should be considered with a caution because of a high number of edible *Russula* species with look-alike poisonous counterparts that often co-occur alongside [63]. Sufficient knowledge about current ecological variation and distribution of *Russula* members can reveal their conservation concern and contribute to protection of endangered habitats where fungal role is still underestimated [64].

## 5. Conclusions

All five species of *Russula* subsect. *Maculatinae* occurring in Pakistani region of western Himalayas are members of at least three independently evolved lineages, but current data do not allow explaining their vicariance, ecological adaptations and evolutionary history. Authors of taxonomic literature frequently do not realise that data about differences in morphology and DNA sequences are often useful only for taxonomists [65]. Ecological data represented by a statistically significant number of observations are an important part of fungal functional traits [66]. The message of this paper is “let’s pay more attention to collection sampling and ecological data to understand the biological identity of species and nature itself”.

## Figures and Tables

**Figure 1 life-11-00662-f001:**
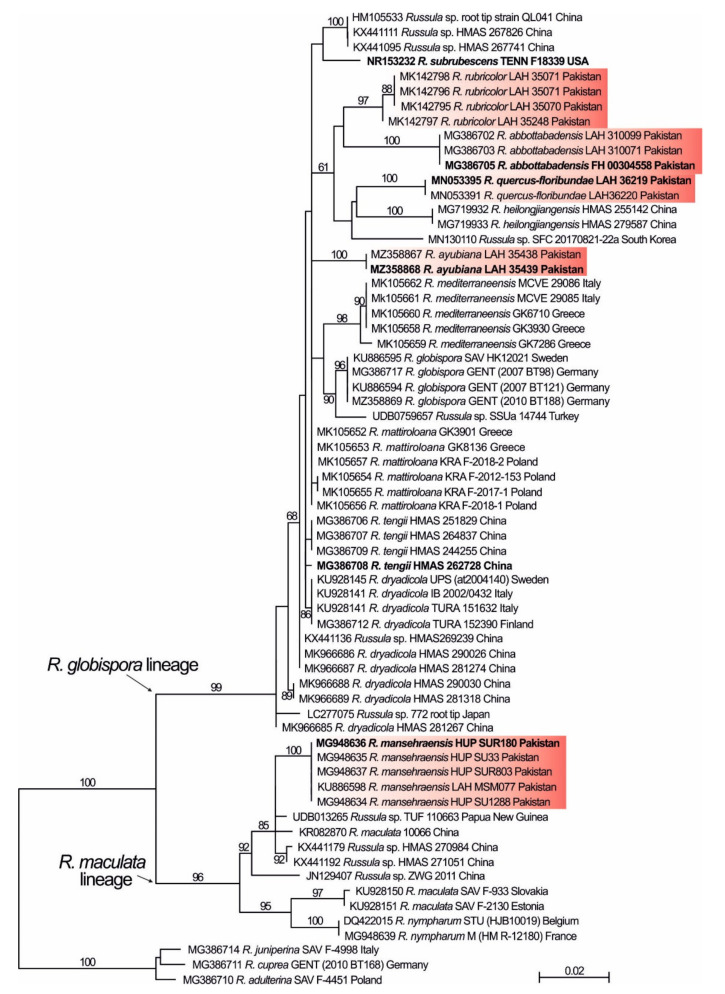
ML tree of the *Russula* subsect. *Maculatinae* based on sequence data of the ITS region. Names of ex-type specimens are in bold. Pakistani species are highlighted by a red background.

**Figure 2 life-11-00662-f002:**
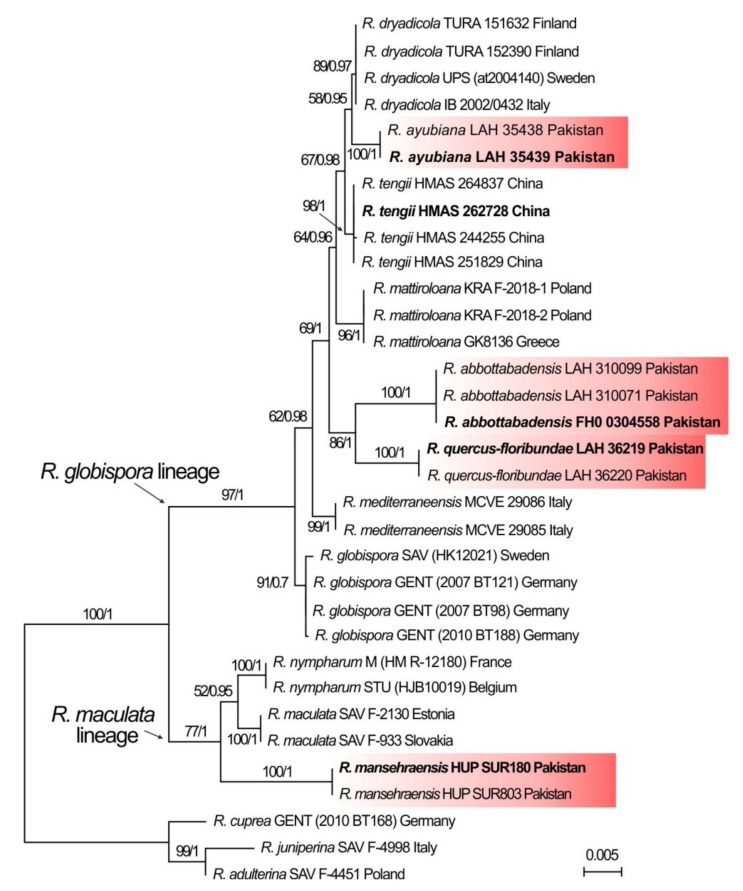
ML tree of *Russula* subsect. *Maculatinae* combined sequence data of nrLSU, mtSSU, *rpb2* and *tef1α*. Pakistani samples are highlighted by a red background. In bold are type collections. The tree is rooted with samples of *Russula* subsect. *Urentes* Maire.

**Figure 3 life-11-00662-f003:**
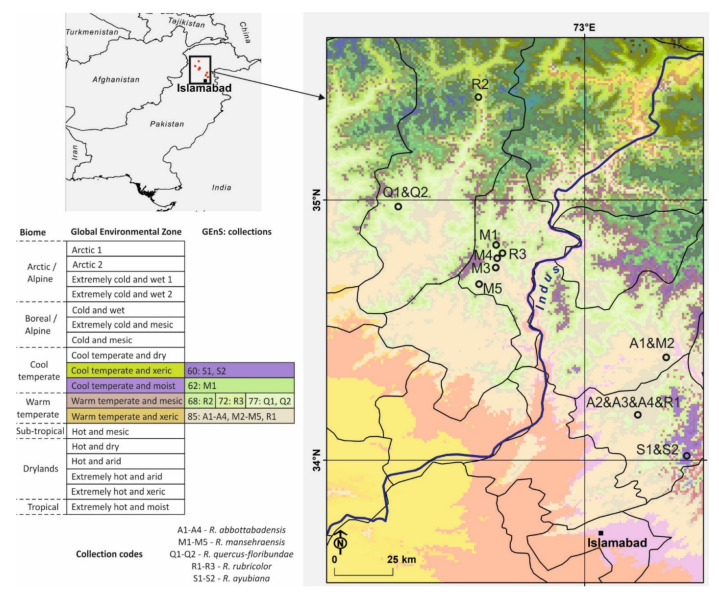
Location of Pakistani *Maculatinae* collecting sites in bioclimatic zones. Global environmental zones with coloured background are positive for the presence and their colour correspond to the bioclimate map modelled by Metzger et al. [41]. Colours of the map background correspond to global environmental strata (GEnS). Collection codes are explained at the **bottom left**.

**Figure 4 life-11-00662-f004:**
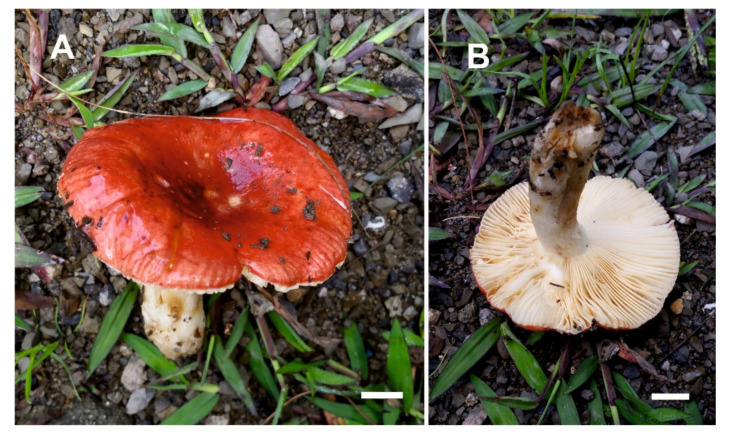
Basidiomata of *Russula ayubiana* in the field (LAH 35439, holotype). (**A**) Pileus surface, (**B**) Hymenium view. Scale bar = 1 cm.

**Figure 5 life-11-00662-f005:**
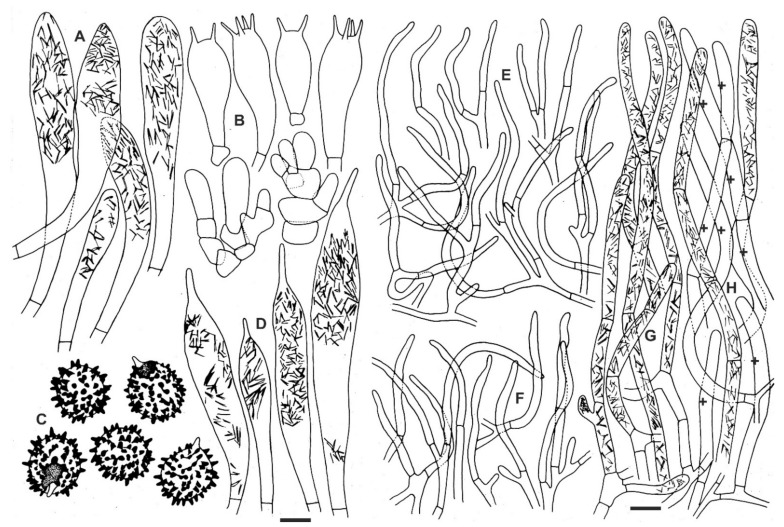
*Russula ayubiana* (LAH 35439, holotype). Elements of the hymenium drawn as seen by light microscopy. (**A**) Hymenial cystidia near the lamellae edges, (**B**) Basidia and basidiola, (**C**) Basidiospores in Melzer’s reagent, (**D**) Hymenial cystidia on the lamellae sides, (**E**) Hyphal terminations near the pileus center, (**F**) Hyphal terminations near the pileus margin, (**G**) Pileocystidia near the pileus margin, (**H**) Pileocystidia near the pileus center. Cystidia with contents as observed in Congo red. Scale bar = 5 μm for basidiospores and =10 μm for all other elements. Drawings by Munazza Kiran.

**Table 1 life-11-00662-t001:** List of all available *Maculatinae* reports from Pakistan with collection details. (T)—type collections.

Species	Coll. Code	Herbarium Acronym (Collection Number)	Collecting Site	Coordinates	Altitude (m a.s.l.)	Associated Trees
*R. abbottabadensis*	A1	FH 00304558	Khyber Pakhtunkhwa: Mansehra, Batrasi	34°23′42.94″ N 73°18′53.03″ E	1109 m	*Pinus roxburghii*
*R. abbottabadensis*	A2	FH 00304589	Khyber Pakhtunkhwa: Abbottabad, Shimla Hill	34°10′27.62″ N 73°12′18.13″ E	1311 m	*Pinus roxburghii*
*R. abbottabadensis*	A3	LAH 310071	Khyber Pakhtunkhwa: Abbottabad, Shimla Hill	34°10′27.62″ N 73°12′18.13″ E	1311 m	*Pinus roxburghii*
*R. abbottabadensis*	A4	LAH 310099	Khyber Pakhtunkhwa: Abbottabad, Shimla Hill	34°10′27.62″ N 73°12′18.13″ E	1311 m	*Pinus roxburghii*
*R. *mansehraensis** (T)	M1	HUP SUR180	Khyber Pakhtunkhwa: Shangla, Puran	34°49′39.1″ N 72°39′37.5″ E	2380 m	*Pinus roxburghii*
*R. mansehraensis*	M2	LAH (MAM 0077) duplicate FH00304559	Khyber Pakhtunkhwa: Mansehra, Batrasi	34°23′42.94″ N 73°18′53.03″ E	1109 m	*Pinus roxburghii*
*R. mansehraensis*	M3	HUP SU1288	Khyber Pakhtunkhwa: Shangla, Bunirwal	34°44′26″ N 72°39′32″ E	1443 m	*Pinus roxburghii*
*R. mansehraensis*	M4	HUP SU33	Khyber Pakhtunkhwa: Shangla, Sanila	34°46′35.6″N 72°39′50.8″E	1264 m	*Pinus roxburghii*
*R. mansehraensis*	M5	HUP-SU803	Khyber Pakhtunkhwa: Shangla, Chowga	34°40′37.8″ N 72°35′42.2″ E	1226 m	*Pinus roxburghii*
*R. quercus-floribundae* (T)	Q1	LAH 36219	Khyber Pakhtunkhwa, Malakand division: Swat, Upper Shawar	34°58′28″ N 72°17′04″ E	1454 m	mixed forest dominated by *Quercus floribunda*
*R. quercus-floribundae*	Q2	LAH 36220	Khyber Pakhtunkhwa, Malakand division: Swat, Upper Shawar	34°58′28″ N 72°17′04″ E	1454 m	mixed forest dominated by *Quercus floribunda*
*R. rubricolor* (T)	R1	LAH 35071	Khyber Pakhtunkhwa: Abbottabad, Shimla Hill	34°10′27.4″ N 73°12′19.6″ E	1311 m	*Cedrus deodara*
*R. rubricolor*	R2	LAH 35070	Khyber Pakhtunkhwa, Malakand division: Swat, Mashkun	35°23′45.17″ N 72°35′32.25″ E	2398 m	*Cedrus deodara*
*R. rubricolor*	R3	LAH 35248	Khyber Pakhtunkhwa, Malakand division: Swat, Towa	34°47′46.0″ N 72°41′05.7″ E	1908 m	*Quercus* sp.
*R. ayubiana* (T)	S1	LAH 35439	Khyber Pakhtunkhwa: Abbottabad, Kuzagali	34°01′02.1″ N 73°23′36.9″ E	2472 m	mixed forest dominated by conifers
*R. ayubiana*	S2	LAH 35438	Khyber Pakhtunkhwa: Abbottabad, Kuzagali	34°01′02.1″ N 73°23′36.9″ E	2472 m	mixed forest dominated by conifers

## Data Availability

Data is contained within the article and Appendix A.

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
