# Peer review of "Description of the Fifth New Species of Russula subsect. Maculatinae from Pakistan Indicates Local Diversity Hotspot of Ectomycorrhizal Fungi in Southwestern Himalayas"

_life, 2021, doi:10.3390/life11070662_

Round 1
Reviewer 1 Report
COMMENTS TO THE MANUSCRIPT “Description of the fifth new species of Russula subsect. Maculatinae from Pakistan indicates local diversity hotspot of ectomycorrhizal fungi in southwestern Himalayas” by Kiran et al.
General comment:
In the submitted manuscript the authors analyze phylogenetic relationships of specimens belonging to the Russula subsect. Maculatinae species and collected in in southwestern Himalayas in northern Pakistan. A novel species, Russula ayubiana sp.nov, within the subsection is reported. Also, integrating the phylogenetic, ecological, and climatic data, the authors stablish hypotheses about the origin and dispersion of species in Maculatinae. They suggest a recent colonization of Maculatinae species of the southwestern Himalayas from different geographical areas an its further adaptation to these microclimatic conditions.
The submitted manuscript is well written, the methodology is properly described, and the results clearly explained and discussed.
The phylogenetic and biogeographic description of new fungal species or knowing ones along different ecosystems and in new geographical areas is of scientific concern to continuously increase the knowledge on the diversity and evolutive patterns of different fungal taxa. By this, the subject of the submitted manuscript is of interest and is suitable to be published in the Life Journal. I show below minor observations and few suggestions to accept the manuscript.
Specific comments:
- In the lines 21-22 of the Abstract the authors states that “…we propose improvements that would facilitate the future non-taxonomic studies of fungi in the region.” This statement is a little bit confusing. Although they are referring biogeography and ecology as subject areas, it may be better to clearly state that they propose the integration of ecological and biogeographical metadata in the phylogenetic analysis of the studied taxon.
- In the line 340 delete the additional dot in “…ecological adaptations, niche specializations, speciation drivers, etc..”
- Joshi et al. (2005. Mycosphere. 3(4): 486–501, doi: 10.5943 /mycosphere/3/4/12) review the diversity of Russula species in the northwestern Himalayas. In this region 50 Russula species have been described so far. No studied species between this region and the southwestern Himalayas in the submitted manuscript are shared. I think that it be useful to discus in the submitted manuscript the lack of coincidence among these Himalayas regions to reinforce their hypothesis on the southwestern Himalayas hotspot for Russula diversification and adaptation to local microclimatic conditions.
- Shimono et al. (2018. Mycoscience. 59(4): 288-293) described 6 Russula species belonging to the different sections collected in the Mt Fuji. This previous study does not report new species and use only the ITS region in the phylogenetic analysis; however, their studied species cluster with high bootstrap values with well described specimens from Europe. Do the authors consider this another example of hot spot speciation of Russula in a restricted geographical area, as you state for southwestern Himalayas?
- The authors propose the new species Russula ayubiana sp.nov. Based on your knowledge on this fungal genus and on the southwestern Himalayas region, do you consider this species needs to be catalogued rare or endangered? Blackwell and Vega (2018. Fungal Ecology. 35: 127-134) states that “In fungal conservation efforts, it is essential not only to discover hidden fungi but also to determine if they are rare or actually endangered.” Can your data be useful to propose conservation and/or management strategies to preserve the diversity of Russula and its probable mycorrhizal associations at this relevant ecological area?
Author Response
We are very grateful for the comments.
We adapted our manuscript following recommendation of the reviewer and here are our answers to specific reviewer s comments:
Specific comments:
- In the lines 21-22 of the Abstract the authors states that “…we propose improvements that would facilitate the future non-taxonomic studies of fungi in the region.” This statement is a little bit confusing. Although they are referring biogeography and ecology as subject areas, it may be better to clearly state that they propose the integration of ecological and biogeographical metadata in the phylogenetic analysis of the studied taxon.
Authors Response: Our intended meaning of the sentence was different, but thank you for the warning about confusing character of the sentence. We focus to taxonomic studies that miss metadata about collections and becomes useless for further non-taxonomic studies. We changed the sentence: “…that would facilitate the integration of ecological and biogeographical metadata from the future taxonomic studies of fungi in the region.”
2. In the line 340 delete the additional dot in “…ecological adaptations, niche specializations, speciation drivers, etc..”
Authors response: done
3. Joshi et al. (2005. Mycosphere. 3(4): 486–501, doi: 10.5943 /mycosphere/3/4/12) review the diversity of Russula species in the northwestern Himalayas. In this region 50 Russula species have been described so far. No studied species between this region and the southwestern Himalayas in the submitted manuscript are shared. I think that it be useful to discus in the submitted manuscript the lack of coincidence among these Himalayas regions to reinforce their hypothesis on the southwestern Himalayas hotspot for Russula diversification and adaptation to local microclimatic conditions.
Authors response: we included this reference in the first paragraph of the Discussion as a proof, that Pakistani Maculatinae members were never detected by any collection or sequence out of the country.
4. Shimono et al. (2018. Mycoscience. 59(4): 288-293) described 6 Russula species belonging to the different sections collected in the Mt Fuji. This previous study does not report new species and use only the ITS region in the phylogenetic analysis; however, their studied species cluster with high bootstrap values with well described specimens from Europe. Do the authors consider this another example of hot spot speciation of Russula in a restricted geographical area, as you state for southwestern Himalayas?
Authors response: we think that this publication rather report species with high dispersal and we give it as example of continuous distribution of Russula members in boreal / arctic areas. We think that southwestern Himilayas are rather example of isolated diversity pool with low share of species diversity with Europe or other distant areas.
5. The authors propose the new species Russula ayubiana sp.nov. Based on your knowledge on this fungal genus and on the southwestern Himalayas region, do you consider this species needs to be catalogued rare or endangered? Blackwell and Vega (2018. Fungal Ecology. 35: 127-134) states that “In fungal conservation efforts, it is essential not only to discover hidden fungi but also to determine if they are rare or actually endangered.” Can your data be useful to propose conservation and/or management strategies to preserve the diversity of Russula and its probable mycorrhizal associations at this relevant ecological area?
Authors response: thank you for this and previous remarks. Sufficient collection data about fungal species can contribute to their commercial use but also they can identify species as of conservation concern. We added this as the last paragraph of the discussion.
Reviewer 2 Report
The manuscript is well designed and the study is outstanding and significant contribution to the field of general mycology.
It is well organized study, well presented, and discussed appropriately.
Please, just add a paragraph in introduction, describing why the Russula genus is important to investigate, not only from the point of diversity, systematics and molecular phylogeny, but also for their use as edible and/or toxic.
Please, also mention the edibility of investigated taxa.
Author Response
We appreciate the positive view of our paper. Our response to reviewer is as follow:
Reviewer 2: Please, just add a paragraph in introduction, describing why the Russula genus is important to investigate, not only from the point of diversity, systematics and molecular phylogeny, but also for their use as edible and/or toxic. Please, also mention the edibility of investigated taxa.
Authors response: We added in the first paragraph of the introduction an information about edibility and toxicity of Russula species. In the last paragraph of the Discussion, we also discuss the knowledge of Russula collected as edible fungus in adjacent regions of Asia and about edibility of Maculatinae generally.